# *Operando* monitoring of ion activities in aqueous batteries with plasmonic fiber-optic sensors

Runlin Wang [1,6], Haozhe Zhang[2,6], Qiyu Liu[2,6], Fu Liu[3,6], Xile Han[1], Xiaoqing Liu[2], Kaiwei Li[1], Gaozhi Xiao[4], Jacques Albert [3], Xihong Lu [2✉] & Tuan Guo [1,5✉]

Understanding ion transport kinetics and electrolyte-electrode interactions at electrode surfaces of batteries in operation is essential to determine their performance and state of health. However, it remains a challenging task to capture in real time the details of surface-localized and rapid ion transport at the microscale. To address this, a promising approach based on an optical fiber plasmonic sensor capable of being inserted near the electrode surface of a working battery to monitor its electrochemical kinetics without disturbing its operation is demonstrated using aqueous Zn-ion batteries as an example. The miniature and chemically inert sensor detects perturbations of surface plasmon waves propagating on its surface to rapidly screen localized electrochemical events on a sub-µm-scale thickness adjacent to the electrode interface. A stable and reproducible correlation between the real-time ion insertions over charge-discharge cycles and the optical plasmon response has been observed and quantified. This new operando measurement tool will provide crucial additional capabilities to battery monitoring methods and help guide the design of better batteries with improved electro-chemistries.

[1] Institute of Photonics Technology, Jinan University, Guangzhou 510632, PR China. [2] The Key Lab of Low-Carbon Chemistry & Energy Conservation of Guangdong Province, School of Chemistry, Sun Yat-Sen University, Guangzhou 510275, PR China. [3] Department of Electronics, Carleton University, Ottawa, ON K1S 5B6, Canada. [4] Advanced Electronics and Photonics Research Center, National Research Council of Canada, Ottawa, ON K1A 0R6, Canada. [5] Southern Marine Science and Engineering Guangdong Laboratory (Zhuhai), Zhuhai 519000, China. [6] These authors contributed equally: Runlin Wang, Haozhe Zhang, Qiyu Liu, Fu Liu. ✉email: luxh6@mail.sysu.edu.cn; tuanguo@jnu.edu.cn

In many ways, renewable energy sources have been identified as key factors towards the replacement of carbon-based energy supply in view of decelerating climate change. However, most of the renewable energy sources, such as wind and solar, do not deliver on demand but rather intermittently and on availability. Therefore, there is a strong impetus on the development of means to store energy temporarily and to deliver it on demand and rechargeable batteries are already used for this purpose but with limited energy storage and power supply capacity. It is therefore imperative to improve battery storage systems so that they can fully replace fossil fuels in all human needs for power utilization, including transportation, heating/cooling, personal electronics, and the power grid in general[1–3]. The new and increasing dependence of society on large- and small-scale rechargeable battery storage systems imposes much higher reliability and performance standards to ensure their safety and their continued usefulness. As a result, and most importantly in view of aging effects that may impact the charge-discharge cycles of these complex electro-chemical systems, it is important to develop new non-invasive monitoring tools for everyday tracking and life cycle management of these systems *in operando* (while in use and without interfering with their operation)[4–8].

The critical factors in battery performance and maximum energy storage capacity are determined by the charging and discharging processes which in turn depend on ion intercalation and de-intercalation kinetics between the electrodes and the electrolytes. In particular, it has long been a challenge in rechargeable battery research to understand the electrochemical reactions occurring during repetitive charges and discharges and how they depend on ion concentrations mobility near the electrode surfaces, and indeed a yet unsolved problem for batteries in operation. The reason for this is that existing methods to quantify charge transport near electrodes are predominantly restricted to inoperative batteries and to time scales that are much longer than the fastest processes occurring real time operation[9,10]. Apart from advances in theoretical models, these methods rely on sophisticated laboratory tools encompassing transmission electron microscopy[11–14], synchrotron-based techniques[15,16], magnetic resonance imaging[17,18], and fluorescence microscopy[19,20]. All of these are obviously impossible to use for routine monitoring of batteries in normal use and there is a dire need for unobtrusive, inexpensive, and reliable devices that could be deployed (at least in large energy storage systems) to monitor the state of health of batteries in real-time and in operation and to relay diagnostic information to system operators. For this to occur, fundamentally new techniques must be developed for implanted multi-parameter sensors that are compatible with the harsh electrolyte environments found within batteries over the course of their expected operating life.

Because of the chemical resistance of silica glass, direct optical detection techniques are promising candidates for the development of in situ sensors. These techniques have shown high sensitivity, fast response time, high spatial resolution and they do not destroy or disturb reactions inside battery cells. Optical measurements techniques recently applied to batteries include internal reflection imaging (TRI), surface plasmon resonance sensing and imaging (SPR), and Raman spectroscopy (SRS). For instance, TRI has been applied to the mapping of the surface activity distribution of an electrode and to quantifying the evolution of oxygen reactions in vanadium flow batteries[21]. SPR technologies are also ideal because they benefit from an increased sensitivity to chemical changes near a metal surface where plasmon waves are excited[22]: they have demonstrated their usefulness in many fields including in situ detection of single proteins[23], and single exosomes[24], and of the charge status of small molecules[25]. On the other hand, SRS directly measures the vibrational frequencies of

chemical bonds in molecules and therefore can identify the electrolyte composition (ions, additives) in the batteries and measure their concentrations. In a recent publication, it has been shown that SRS imaging can follow quantitatively the fast evolution of the ion concentration at the electrode surface of a Li-ion battery and that the concentration was related to the growth of Li dendrites[26]. However, all these experiments were carried out in laboratory settings using bulk optic components and free-space optical propagation. They were therefore incompatible with in situ measurements within batteries and even less in working batteries. It is imperative to develop miniature and remotely operated solutions along the same lines and to demonstrate that similar quantitative information can be acquired.

Optical fiber sensors represent a very promising approach towards such solutions, as they possess the required form factor, electrical immunity and remote operation ability. They have been demonstrated in many other fields as multiparameter sensing tools with excellent temporal and spatial resolution. In particular, fiber Bragg grating (FBG) devices are used to measure factors such as the local temperature ($T$), pressure ($P$), and strain ($\varepsilon$) by remote optical power spectrum measurements of the light reflected from the mm-sized sensor at the measurement point[27–31]. Implanted FBG sensors have already been shown to measure cell temperature[32,33], to provide state of charge and state of health estimations[34–36], and to differentiate thermal and strain effects[37,38]. While these measurements are helpful in correlating physical state variables with battery performance, they provide no information on the critical chemical parameters. In a first step to address this, an SRS multifiber probe was used by Yamanaka and his team to demonstrate the feasibility of in situ measurements of ion concentrations near a Li-ion battery electrode surface and to indentify permeability as a key factor in improving battery[39]. It was only much more recently that a team led by Tarascon managed to decode some chemical and thermal processes in commercial Na(Li)-ion cells, in operando, with structured FBG devices[40]. These preliminary findings demonstrate the potential of fiber optic approaches for the development of scalable solutions for improving battery thermal management (and safety) in parallel with optimization of electrolyte-electrode compositions for longer and more stable charge-discharge cycling performance. In summary, research is now ready to develop more advanced tools which can measure physical, chemical and electrochemical parameters simultaneously, in parallel or combined, with high temporal and spatial resolution inside working batteries. This will initially advance our understanding of the underlying electrochemical processes but it also should lead to practical systems that can be deployed for monitoring installed systems and inform maintenance and replacement schedules.

Herein we make progress by demonstrating the feasibility of incorporating an electrochemical SPR fiber-optic sensor into batteries for in situ detection of chemical and electrochemical events, without perturbing battery operation. The sensor adopted is a tilted fiber Bragg grating (TFBG) imprinted in a commercial single-mode fiber and coated with a nanoscale gold film for the high-efficiency excitation of surface plasmon polaritons. The phase velocity and attenuation distance of the SPR wave on the fiber sensor surface located adjacent to the electrode are modified by small and local ion transport differences, leading to changes in the position and shape the TFBG-SPR spectral features measured in reflection. In contrast to existing techniques (such as cyclic voltammetry (CV)) that rely on a "bulk" battery-averaged estimation, our sensor aims at directly quantifying the amount of ions transport and intercalation directly on the electrode surface. As a proof of concept, we initially focused on aqueous Zn-ion rechargeable batteries before generalizing our approach to other electrolytes and battery chemistries. The function of promoting

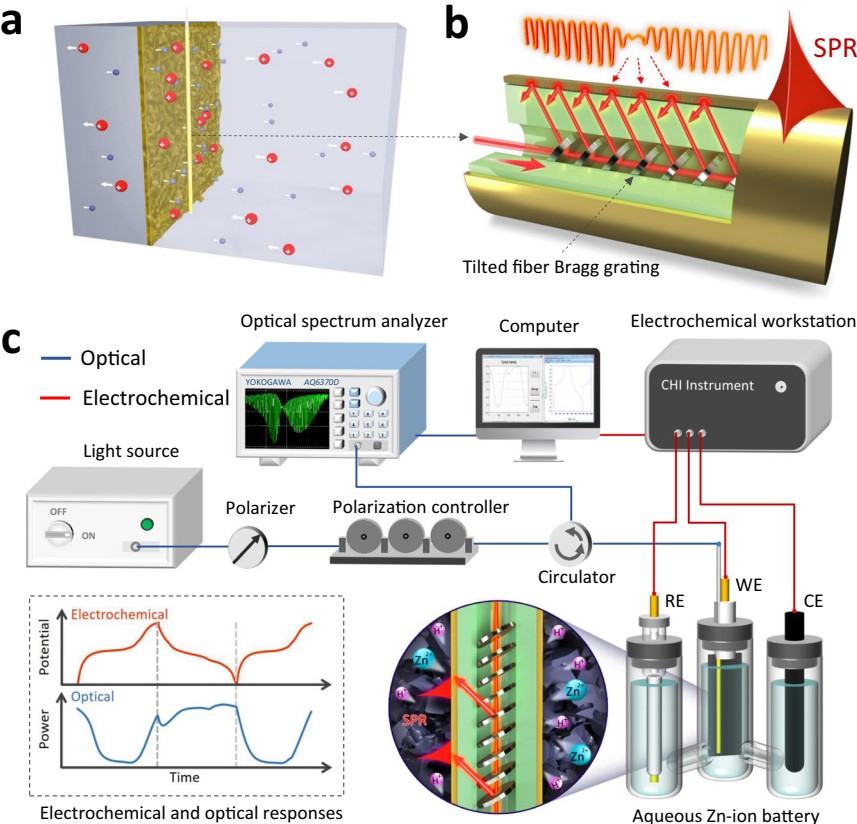

**Fig. 1 Concept of plasmonic optical fiber sensing inside the battery. a** Schematic of a gold-coated fiber-optic sensor closely attached electrolyte-electrode interface of an aqueous battery for in situ detection interfacial ion concentrations and ion transport activities; **b** Sketch of the configuration of a plasmonic fiber-optic sensor. A tilted fiber Bragg grating (1 cm in length) imprinted in a commercial single-mode fiber (125 μm in diameter) and coated with a gold film (50 nm in thickness) for high-efficiency surface-plasmon-resonance (SPR) excitation; **c** Experimental setup of a plasmonic fiber-optic sensing system and an electrochemical calibration system for Zn-ion battery measurement. RE: reference electrode, WE: working electrode, CE: counter electrode. The dash-dot-line inset presents the voltage of the battery (top) and corresponding optical output of the plasmonic fiber sensor (bottom) simultaneously.

$H^+$ intercalation kinetics in $MnO_2$ cathodes by the PEDOT layer is observed and analyzed from the in-situ kinetics study results. This simple-to-implement method fills an important gap in current battery chemical and electrochemical monitoring in real time, offers a scalable solution for screening surface (sub-μm-scale) chemical events adjacent to the electrode interface with high accuracy, rapidly identify the detailed information of interfacial processes and can be integrated as part of existing battery components for *in operando* monitoring.

## Results

***Operando* monitoring electrochemical activities at electrolyte-electrode interface.** As discussed above, in situ and continuous monitoring the electrochemical reactions and interfacial kinetics in the proximity of electrode surfaces is challenging. We try to address this long-lasting question by implanting our plasmonic fiber-optic sensor into the battery and closely attaching it to the surface of the electrode, as shown in Fig. 1a in principle and Fig. S1 in experiments. Taking electroplated $MnO_2$ electrodes (denoted as MO) as an example, the attached plasmonic fiber-optic sensor does not affect the electrochemical performance of the electrode (Fig. S2). Contrary to the case for standard fiber Bragg gratings, the planes of the refractive index modulation of TFBGs were written with a predefined tilt relative to the longitudinal axis of the fiber, as shown in Fig. 1b. Differing from the traditional SPR based on bulk prism, a tilted fiber grating assisted SPR sensor provides "two hybridized resonances" as Fig. S3 shows. By tilting the grating fringes, the core-to-cladding

resonant mechanism has the effect of providing a high-density narrowband spectral comb (tens to hundreds of narrowband cladding resonances with individual bandwidth ~0.1 nm and Q-factor at the level of $10^4$) that overlap with the broader absorption (more than 10 nm wide in the best of cases) of the surface Plasmon. Such hybridization of the grating resonances with that of the SPR provides a unique tool to measure small shifts of the SPR (due to RI changes) with high accuracy by measuring amplitude changes of the grating resonance located on the edge of the SPR as it shifts, offering the limit of detection for refractive index in the range of $10^{-6}$ to $10^{-8}$ [41,42]. More importantly, compared to the conventional SPR based on ultraviolet–visible light, TFBG-assisted fiber-optic SPR works in the near infrared and therefore provides a much longer penetration depth together with propagation length, which can be enhanced to more than 500 nm and 1 mm (at the wavelength of 1550 nm), respectively [43]. Such propagation length cover the most active electron transfer [44] and ion transport [45] area over the electrode surface. Any slight change in ions distribution around the electrode can be directly monitored by reading the changes of the SPR spectrum of the sensor. The tilt angle of grating is an important parameter that determines which set of cladding modes is excited. The selection of the optimum tilt angle of the grating is determined by the refractive index of the electrolyte used in the Zn-ion batteries. The maximum sensitivity is obtained when the cladding modes have effective indices close to the index of the electrolyte which has been measured to be 1.33, with fluctuations of less than 0.02 during charging/discharging cycles.

For a grating with a Bragg wavelength near 1620 nm, cladding modes with effective indices near 1.33 occur in the spectral region between 1560 and 1570 nm.

In order to maximize the amplitudes of the cladding modes at these wavelengths, the tilt angle was selected to be 12 degree based on simulations of TFBG spectra (Fig. S4). Experimental results confirm that the maximum of the envelope of resonance amplitudes moves towards shorter wavelengths as the tilt angle increases and that the SPR location falls in the correct spectral window. Finally, different from nanoparticles-assisted localized SPR[46], such long range SPR sensor is fabricated in a grating-assisted commercially available "normal" singe mode fiber, which therefore ensures the reliability and feasibility (with no structural modifications) together with low cost and process availability (the well-established phase masks grating inscription with standard batch-coating processes) to ensure that the devices produced are identical. It provides sensing inside batteries with highly stable and reproducible measurement.

Figure 1c shows the all-fiber-coupled electrochemical SPR fiber-optic sensing system (see details in the section of Method 4.1 and Fig. S1). The optical fiber sensing probe was tightly attached to the surface of working electrode while a fixing device. This configuration also ensures strain-free operation of the sensor to eliminate the effect of cross sensitivity to strain of the higher-order cladding and plasmonic modes when the sensor is fixed at one end. The three-electrode system was driven by an electrochemical workstation. Both the optical signal and electrochemical signal were recorded and analyzed by a computer in real time. The dash-dot-line inset presents the voltage of the battery (top) and corresponding optical output of the SPR fiber sensor (bottom) simultaneously. We will discuss later how the surface-localized and fast changing charge transport kinetics near the electrode surface can be extracted from this tiny fiber-optic sensor during the battery operation.

**Spotting the ion insertion at electrode surface**. The main advantage of TFBGs over conventional SPR sensors is best appreciated in Fig. 2a where the attenuation band of the TFBG spectrum due to the SPR effect is highlighted in red. The spectral bandwidth of the SPR envelope is at least of the order of 5–20 nm for the best cases (including for theoretical calculations for the seminal case of the Kretschmann-Raether configuration with the same thickness of gold on a silica glass surface[41]). This broad bandwidth limits the accuracy with which the SPR wavelength can be tracked as it shifts. So instead of attempting to follow the displacements of the broad SPR envelope, an alternative approach is developed based on the very dense spectral distribution of cladding mode resonances. This density always allows the identification of a plasmonic resonance, located on the short wavelength side of the SPR envelope and identified by the red asterisk "*". As the attenuation band of the SPR shifts, the amplitude of the plasmonic resonance changes and allows an indirect, but precise, quantitative measurement of very small SPR changes (as shown in Fig. 2d for instance and analyzed in Fig. S5 comprehensively).

Figure 2b, c shows the numerical simulation result of the SPR energy distribution and how the light transfer from the fiber core to the surface metal layer. Clearly, the SPR significantly enhances the localization of the optical mode power near the fiber surface. Simulation results show that it localizes up to 70% of the TM/EH fields of the cladding mode power in the external medium, while the mode power fraction in the evanescent field of the non-plasmonic modes of the same TFBG does not exceed 2–5%[43]. Meanwhile, it should be emphasized that the plasmonic power distribution is not uniform around the fiber circumference.

Because the inherent coupling properties of the TFBG, the cladding modes have most of their power distributed in two lobes diametrically oriented along the tilt plane. Figure 2d presents the zoomed spectra of SPR modulated cladding mode resonances when the battery is charging. Only a few specific cladding modes (around 1566 nm) are modulated with a stable and repeatable matter, while all the others are kept unchanged. We name this part of wavelength as the "electrochemical sensing" region, just like an "opened window" to see the media around the fiber.

The experimental measurement results of the sensing system in repeated galvanostatic charge/discharge (GCD) tests are shown in Fig. 2e. We clearly find that the response of the optical signal (blue curve) obtained from the optical fiber sensor shows a very stable and correlated relationship with that of electrochemical outputs (red curve). A very interested point is that during the process of battery discharge, there always shows a two-stages change, as the arrow marked. Further study shows that this is associated with the two-step ion insertion process ($H^+$ and $Zn^{2+}$ intercalations).

Moreover, another most important feature of the TFBG spectrum is the presence of the core mode resonance that is immune to the external medium. Therefore, it can advantageously be used to de-correlate unwanted fluctuations of temperature and light power level effects from the sensor response. In above experiment, during all the charging and discharging process, the core mode at the wavelength of 1615 nm remains fixed, shown in the Fig. S6. If the core mode does change (in power or wavelength), in addition to providing a local temperature measurement, it can be used to correct the electrochemical sensing part of the spectrum (SPR modes) for local temperature changes by using the relative wavelength shift of cladding mode to core mode. Because both of they are showing the same temperature sensitivity, shown in the Fig. S7. In practical application inside batteries, the temperature variation will be compensated in real time during the data collection of both cladding mode and core mode, by a data processing correction of the cladding mode shifts from that of the core mode shift. For different concentrations of $H^+$ and $Zn^{2+}$ measurements, the SPR mode shows linear wavelength responses, while the core mode is very stable during this process, further indicating that the effects of temperature change and light source fluctuations are negligible (Fig. S8).

The electrochemical and optical analysis of the ion intercalation at $MnO_2$ cathode were characterized by the three-electrode system. The $MnO_2$ was prepared on carbon fiber paper by a simple electroplating method, and its corresponding scanning electron microscopy (SEM), X-ray diffraction (XRD), and X-ray photoelectron spectroscopy (XPS) characterization results are shown in Fig. S9 to S11. After electroplating, a dense layer is covered on the carbon fiber paper, while Mn and O elements are uniformly distributed on the entire substrate. All of the diffraction peaks in the XRD pattern are well assigned to that of the $MnO_2$ (JCPDS#30-0820), again confirming the successful preparation of MO cathode[47]. Moreover, the XPS survey spectrum of the MO sample further demonstrates that only Mn, O, and C are present and that Mn exists in the form of $Mn^{4+}$. The GCD test was applied to characterize the electrochemical performance of MO cathode (Fig. S12). Specifically, the charge and discharge curves at various current densities both possess the obvious redox plateaus, especially at $0.5\,mA\,cm^{-2}$, where two discharge plateaus can be clearly observed (arrow marked in Fig. S12), revealing a two-step ion intercalation process. To further verify the correlation between electrochemical behavior and optical signal, the curves of GCD test, SPR power and differential of light power are exhibited in Fig. 3a. The differential of light power (blue curve in Fig. 3a) is obtained by taking the derivative of the SPR power level with respect to time, which represents the rate of change of the optical power. It shown that the changes of electrochemical curves

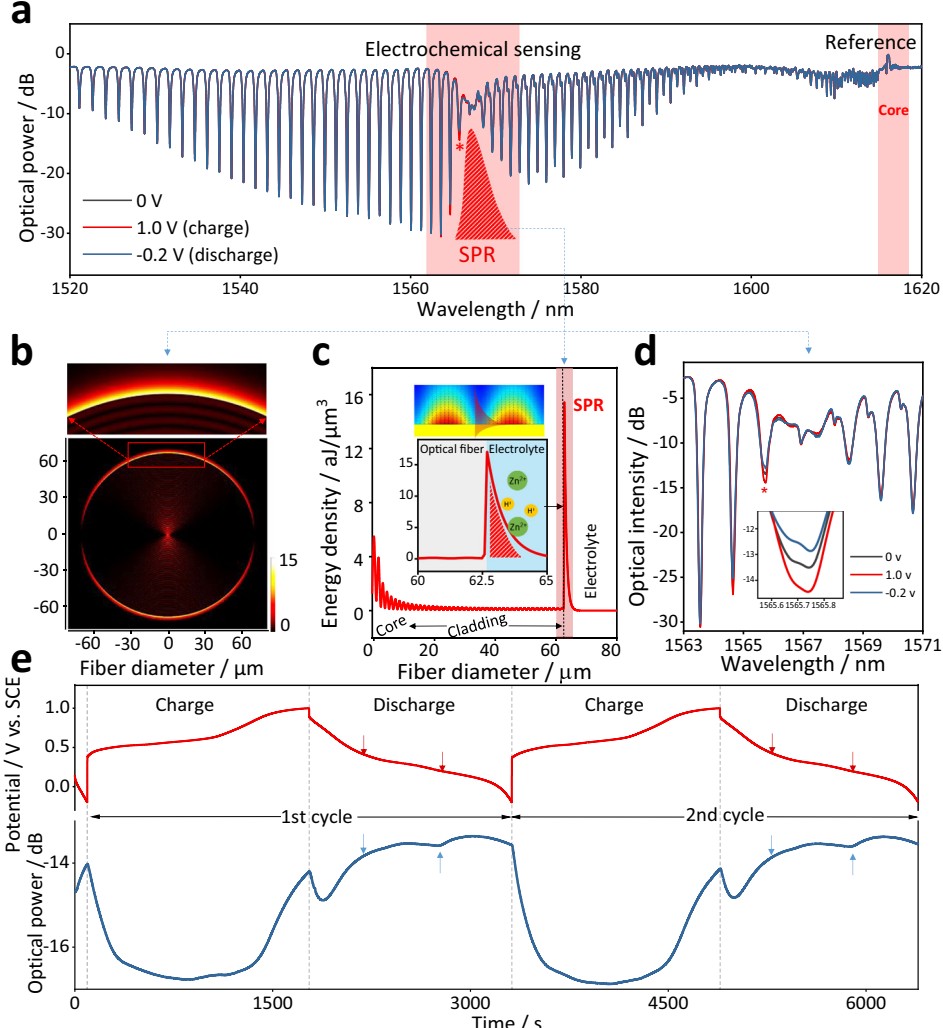

**Fig. 2 Electrochemical surface-plasmon-resonance sensing principle and experimental demonstration with a gold-coated TFBG optical fiber sensor at electrolyte-electrode interface. a** Reflection spectral response of a mirror-ended TFBG optical fiber (tilt angle of 12°) for the different electronic polarization states of the electrode; the spectral change of the most sensitive SPR-coupled cladding mode (marked by a red asterisk "*") is used for electrochemical monitoring; other cladding mode resonances at longer wavelengths and the core mode resonance near 1610 nm can be used for reference. **b, c** Simulated electric mode field profiles of high efficiency SPR over fiber surface excited by P-polarized incident light in the fiber core. The transfer of energy from fiber core to surface plasmon shows up as a bright ring around the fiber boundary (especially visible in the enlarged image). **d** Detailed view of the spectral change of the SPR-coupled cladding mode used for highly sensitive electrochemical measurement. **e** Stable and reproducible correlation between the time-resolved voltage of electrochemical station (red) and the optical power level of the implanted fiber sensor (blue) in charging/discharging cycling tests of a Zn-ion battery.

are highly consistent with the optical results. And most importantly, we find that it shows a stable and reproducible correlation with ion transfer rate. At the time corresponding to the two discharging platforms of 0.62 V (red star) and 0.18 V (blue star), the SPR power decreases, while the differential of light power curve reaches the peak. This is because ions quickly transfer and intercalate cathode material during discharge, thus reducing the ion concentration at the electrode-electrolyte interface. And optical curves flatten out towards the end of the discharge. While starting charging (black star), the optical signal decreased sharply and a peak was observed in d(dB)/dt curve. Since both the electrochemical signal and optical signal are functions of time, the change rate of the optical signal can be mapped to the voltage (Fig. S13) as a P'/V relationship curve. Figure 3b presents the P'/V curves of the first five charging and discharging cycles of MO cathode, which is similar in shape to the CV curve. The 1st cycle is irreversible due to the change in crystal structure, which is common in MnO$_2$ electrode materials[48]. The

curve gradually levels off after the 2nd cycle, indicating a reversible redox reaction with ion intercalation/deintercalation. Furthermore, it can also observe a charging plateau and two discharging plateaus.

For better understanding, two groups of aqueous electrolytes were set to compare the difference between H$^+$/Zn$^{2+}$ two-step intercalation and only H$^+$ intercalation: one is 1 M ZnSO$_4$ + 0.4 M MnSO$_4$, the other is 0.4 M MnSO$_4$, with a pH adjusted the same level as the other. Focusing on the ion intercalation process, Fig. 4a, b shows the discharge curves at the current density of 1 mA cm$^{-2}$ and their corresponding SPR power level change. Apparently, the SPR signal provides more stable signals during each intercalation cycle compared to the electrochemical signal. Unlike voltage change, the SPR power level change is due to the local ion concentration decrease at the diffusion layer caused by H$^+$/Zn$^{2+}$ intercalation. Figure 4c is obtained by taking the derivative of the SPR power level with respect to time, which represents the rate of change of the optical power. The two peaks thus indicate the occurrence of the

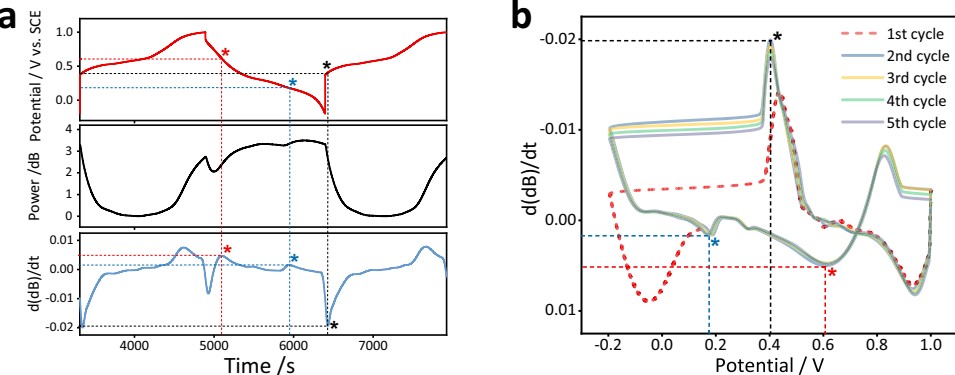

**Fig. 3 Correlation between the differential optical power evolution d(dB)/dt and the temporal potential. a** Real time response of GCD curve (red), optical power level in dB (black), and d(dB)/dt (blue). **b** The relationship curve between the optical signal d(dB)/dt and the potential: each trace follows the simultaneous values of the two parameters as a function of time.

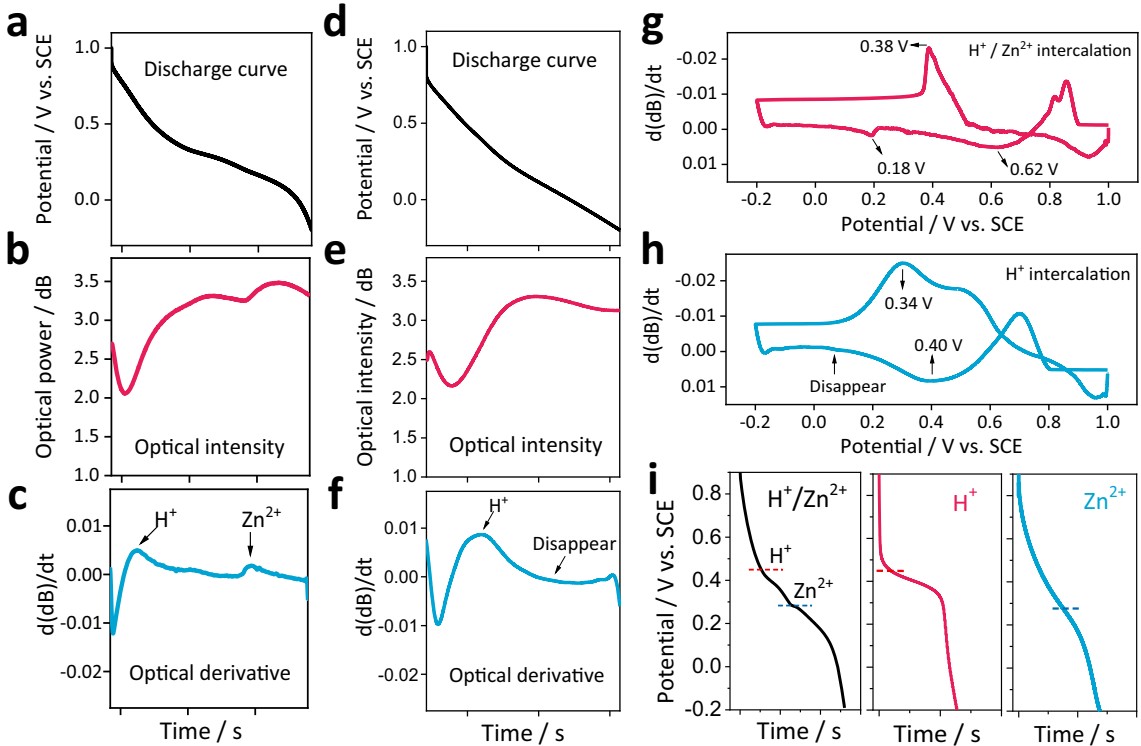

**Fig. 4 Electrochemical curve (black), SPR power curve (red) and differential of light power (blue) *versus* the discharge stages. a–c** 1 M ZnSO$_4$ + 0.4 M MnSO$_4$ aqueous electrolyte, **d–f** 0.4 M MnSO$_4$ aqueous electrolyte. The P'/V relationship curve of the MO electrode during discharging at **g** 1 M ZnSO$_4$ + 0.4 M MnSO$_4$ and **h** 0.4 M MnSO$_4$ aqueous electrolyte. **i** Galvanostatic discharge curve of the MO electrode at 1 M ZnSO$_4$ + 0.4 M MnSO$_4$ (black), 0.4 M MnSO$_4$ (red) and 1 M ZnSO$_4$ (blue) aqueous electrolyte, respectively.

maxima in the rate of change of ion concentration induced by H$^+$/Zn$^{2+}$ intercalation. Figure 4d–f show the results of control group in which the removal of ZnSO$_4$ from aqueous electrolyte ensures that only H$^+$ intercalation occurs under this condition. The results confirm that only one platform/peak exist in this case and that it occurs early in the process which indicates that the intercalation of H$^+$ is earlier than that of Zn$^{2+}$ in the first experiment, which is consistent with previous results[49,50].

Since both the electrochemical signal and optical signal are functions of time, the change rate of the optical signal can be mapped to the voltage (Fig. S9) as a P'/V relationship curve. This curve can intuitively show the rate of change of ion concentration

near the electrode at different potentials in the charging and discharging stages. Figure 4g, h presents the P'/V curves for the two types of electrolytes. Two characteristic potentials are identified for H$^+$/Zn$^{2+}$ intercalation condition in the discharge stage, and there is only one peak for the H$^+$ intercalation. Moreover, the relationship curve can directly reflect the evolution rate of ion concentration at different potentials. Similar to the redox peak current of the CV curve, the peak rate of ion concentration change near the electrode is generated at specific potentials. It can be seen that the peak of H$^+$ intercalation appears at a higher potential (0.62 V and 0.4 V, respectively), while that of Zn$^{2+}$ is lower (0.18 V). By comparing with the

electrochemical curves of the low current density GCD test (the two stages can be clearly observed only at a lower current density) under the three electrolyte conditions presented in Fig. 4i, it can be found that the regularity and trend of the optical signal and the electrochemical signal are basically the same, but more importantly that the H$^+$ intercalation stage occurs at higher potential than that of the Zn$^{2+}$ intercalation stage, thereby confirming the usefulness of the optical signal.

Through the P'/V mapping curves discussed above, the potential polarization characteristics of different materials are further verified. A thin PEDOT layer is coated on the MO and this MO@PEDOT electrode should exhibit boosted electrochemical performance, both in terms of capacity and cycle stability (Figs. S14–S16). Figure S17a compares the optical results of P'/V curves for the MO and MO@PEDOT electrodes. It is obvious that the characteristic potential of the two materials has shifted. Specifically, the characteristic peak potential of MO@PEDOT material decreases from 0.49 V to 0.38 V compared with that of MO material in the charging stage. At the discharge stage, the potential of H$^+$/Zn$^{2+}$ increased from 0.48 V/0.11 V to 0.62 V/0.17 V. To sum up, the MO@PEDOT electrode has lower charging potential and higher discharge point, demonstrating its improved battery performance. In addition, the peak value of MO@PEDOT is higher than that of MO, proving its faster ion transfer rate, that is, more amount of ion intercalation. The CV curves in Fig. S14b show the two materials also possess two distinguishable peaks in the discharge process and a higher current is detected on MO@PEDOT due to more amount of electron transfer. All above results are consistent with those obtained by optical sensing, which further proves the two-step ion insertion mechanism and improved battery performance of MO@PEDOT electrode. Note that, the information provided by CV and P'/V test is different in details as they record different response parameters corresponding to various aspects of the electrochemical system. The P'/V curve represents the change rate of ionic concentration at electrolyte-electrode interface, which reflects ion kinetics and electrode interactions. In comparison, the CV curve presents the current characteristics related to the redox potential, current intensity, and charge/discharge reaction details for electrode performance quantification, such as the calculated specific capacity of MO (65 mA h g$^{-1}$) and MO@PEDOT (174 mA h g$^{-1}$) electrodes. The two processes do not reach maximum values at the same time and the peak of the P'/V curve occurs before the CV curve. In battery testing, the peaks of CV curve indicate the most violent redox reaction inside the electrode while the peaks of P'/V curve reveal the fastest ion transport at the diffusion layer. In addition to the GCD and CV tests, this optical sensing is also applicable to in situ detection of ion kinetics near the electrode surface in other electrochemical models, such as potential step chronoamperometry (PSCA) test (see Supplementary information for details, Fig. S18). Therefore, traditional electrochemical techniques and optic-fiber sensing are two complementary methods for the evaluation of energy storage performance of aqueous batteries and in-depth exploration of their electrochemical mechanism.

**Quantifying the ion insertion kinetics**. The intercalation mechanism and polarization properties of H$^+$/Zn$^{2+}$ ions are verified by the joint analysis of optical signals and electrical signals above, and several important information are obtained: (1) The evolution of SPR power corresponds to the change of ionic concentration, and the two ascending plateaus at the discharge stage correspond to the discontinuous change of ionic concentration in the diffusion layer caused by the intercalation of H$^+$ and Zn$^{2+}$. (2) The differential curve of SPR power d(dB)/dt

corresponds to the rate of change of the ionic concentration in real time, and the two characteristic peaks at the discharge stage correspond to the maximum evolution rate of two kinds of ions. (3) The mapping curve of the differential SPR power with voltage corresponds to the change rate of ion concentration at different potentials. Based on the above conclusions, we further analyzed the effects of materials on the intercalation kinetics of the two ions by a semi-quantitative method. In order to eliminate the possible differences in optical response caused by electrochemical parameters, we first normalized the optical signal according to the maximum variation of SPR power change during the whole charge-discharge cycle, and then differentiated the normalized results to obtain the normalized rate curve. Figure 5a, b is the normalized curve of MO and MO@PEDOT electrodes, respectively (the full curve is shown in Fig. S19). It can be found that the H$^+$ intercalation process of MO electrode causes the normalized SPR power plateau to change by 0.18 but by 0.06 for the intercalation of Zn$^{2+}$. For the MO@PEDOT electrode, the change in the H$^+$ intercalation plateau is obviously larger, reaching 0.36, while the Zn$^{2+}$ platfeau is similar at 0.07. Further observation of the normalized optical rate curves of the two electrodes shows that the change rate of MO@PEDOT electrode during the H$^+$ intercalation process is larger, while the change rate caused by Zn$^{2+}$ intercalation of the two materials is almost the same. The comparison results are shown in Fig. 5c, which clearly demonstrates that the performance optimization of PEDOT material comes from the improvement of H$^+$ diffusion kinetics. In addition, according to the electrochemical impedance spectroscopy (EIS) of each potential during the discharge process (Fig. S20), the charge transfer resistances ($R_{ct}$) of the MO@PEDOT electrode (ranging from 2.2 to 11.6 ohm) are lower than that of MO electrode (ranging from 19.1 to 26.5 ohm), which further demonstrates that the introduction of PEDOT layer strengthens the H$^+$ diffusion (Fig. 5d). This may be attributed to the abundant oxygen and sulfur functional groups in PEDOT which promote the adsorption of H$^+$[51,52]. So far, we have analyzed the influence of materials on the difference of the two ions embedding kinetics by means of optical signal semi-quantization. However, it should be noted here that the normalized value of SPR power does not have specific meaning, but only provides a reference for comparative analysis. Even so, the in situ monitoring of fiber optics offers further insights into battery analysis.

## Discussion

Much published research has demonstrated that the introduction of a PEDOT layer can boost the electrochemical performance of cathodes for Zn-ion batteries, which is mainly attributed to the improvement of the electron conductivity. Actually, the PEDOT layer may contribute to the energy storage, but it remains challenging to directly prove the mechanism of this improvement due to the limitation of available detection techniques[53,54]. Therefore, our electrochemical SPR optical fiber sensor, which can realize in situ and operando optical detection of ion concentrations and ion transport activities at the electrode-electrolyte interface is of great significance. The TFBG-SPR response enables us to elucidate the fundamental cause of the improved performance of MO@PEDOT cathode (Fig. 5e). First, the two-step ion intercalation mechanism of the MnO$_2$ cathode during the discharge is intuitively demonstrated along with the identification of the first plateau as the H$^+$ intercalation stage and the second plateau as the Zn$^{2+}$ intercalation stage. Subsequently, the ion concentration at the electrolyte-electrode interface of the MO@PEDOT electrode changes much more than that of the MO electrode during the H$^+$ intercalation process, which directly proves that the PEDOT layer improves the overall H$^+$ diffusion kinetics of the

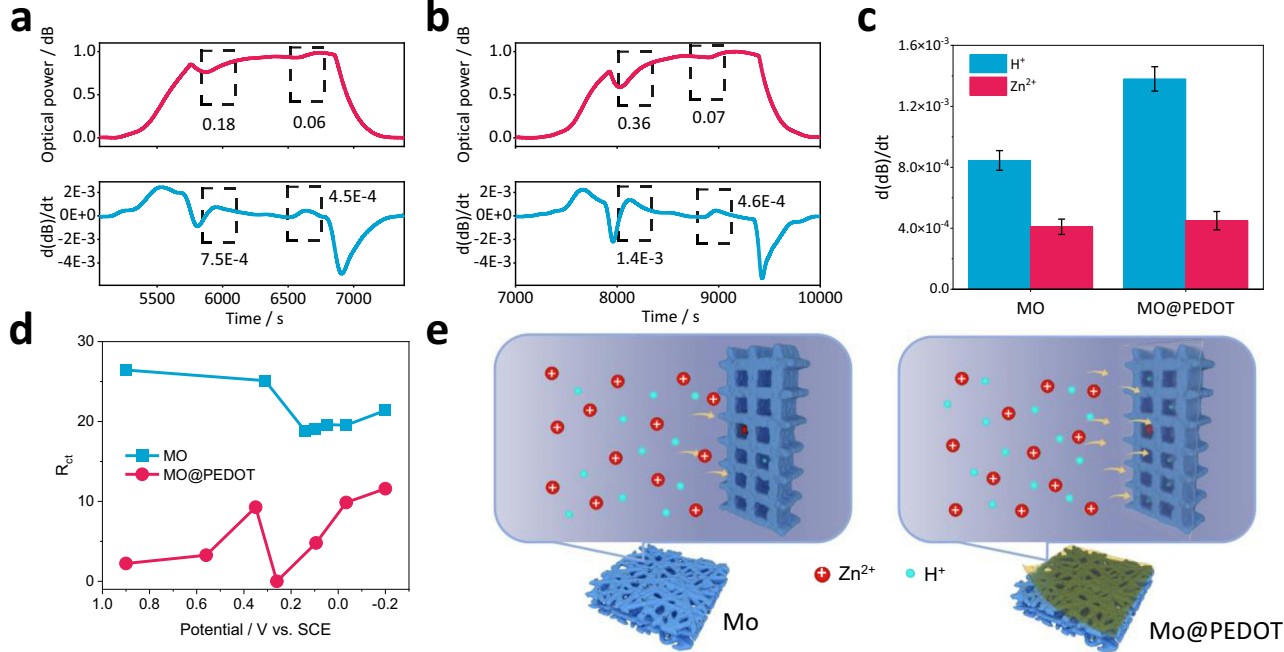

**Fig. 5 Investigation of kinetics in H+ and Zn2+ intercalation.** SPR power curve (red) and time derivative of light power (blue) at the discharge stage of **a** MO and **b** MO@PEDOT electrodes. **c** The change rate of normalized optical rate curves and **d** $R_{ct}$ at different voltages of MO and MO@PEDOT electrodes during discharge process. Vertical error bars show the standard deviation from five continuous charging/discharging cycles. **e** The mechanism of PEDOT layer promoting H+ diffusion kinetics. The abundance of O and S functional groups in the PEDOT layer promotes ion diffusion kinetics, especially H+.

bulk $MnO_2$. This can be interpreted as the contribution from the abundant O and S functional groups on the PEDOT surface, which favors the H+ insertion and diffusion. As result, the H+ intercalation process of MO@PEDOT cathode is more efficient, and thus it exhibits a higher capacity.

In summary, we have successfully demonstrated the feasibility of an electrochemical SPR optical fiber sensor for in situ detection of ion kinetics at the electrolyte-electrode interface without interfering with battery operation. As to the applications of this device, the two-step ion intercalation process of $MnO_2$ as an aqueous Zn-ion battery electrode has been clearly proved by the change of optical signal during discharge process. Moreover, a semi-quantitative method based on the normalized optical rate curves has been developed to further analyze the different ion kinetics between the two electrode materials. Using this method, it is further demonstrated that a PEDOT layer coating on $MnO_2$ can significantly optimize the H+ diffusion kinetics of the electrode, thus obtaining a more satisfactory electrochemical performance.

Moreover, in many circumstances the use of full-scale optical power spectrum measurements with benchtop equipment may be impractical or even impossible, such as routine performance checks of batteries in electric-powered vehicles for instance. In such cases, a portable system consisting of a fixed wavelength (or slightly tunable) semiconductor light source, such as a VCSEL for instance, with a photodetector can be used (there are commercial versions of such systems that are beginning to appear on the market). This only requires that all sensors to be measured have similar spectral characteristics, which can be achieved through mass production methods. The interrogation of sensors is then based on edge filtering where the wavelength of the light source is located near the middle of the of the edge of the most sensitive SPR mode and where transmitted power level fluctuations through the sensor due to SPR shifts are measured with the photodiode, possibly referenced to a direct measurement of the incident light source power (Fig. S21).

Last but not least, the complementary information provided by optical sensing and conventional electrochemical techniques will shine more light on the exploration of ion kinetics and electrochemical mechanism, which may fill an important gap in current battery monitoring methods and will guide the design of new materials, and will optimize existing and new electro-chemistries.

## Methods

**Sensing system.** The *operando* sensing system for monitoring the kinetics of H+ and Zn2+ during the electrochemical process consists of two major subsystems (Fig. 1): an optical system used for in situ detection and a three-electrode system used for calibration. The optical system is made up of a broadband light source with bandwidth from 1250 to 1650 nm, a polarizer, a polarization controller, an optical fiber circulator, a plasmonic fiber-optic sensing probe and an optical spectrum analyzer. They are connected sequentially via standard fiber optic connectors. A manganese dioxide cathode as the working electrode was immersed in the electrolyte, and the optical fiber sensing probe was tightly attached to its surface. A graphite rod and a calomel electrode were set as the counter electrode and the reference electrode. The three-electrode system was driven by an electrochemical workstation. Both the optical signal and electrochemical signal were recorded and analyzed by a computer in real time.

**Fabrication of plasmonic fiber-optic sensor.** The TFBGs were fabricated in a 125-μm-diameter photo-sensitive glass fiber that has a single mode core at the wavelengths used, between 1500 and 1620 nm (FIBERCORE PS1250/1500). For the grating inscription, intense light with a wavelength of 193 nm from a Bragg Star Industrial excimer laser (Coherent, Inc.) was diffracted into a periodic interference pattern by a phase mask with a period of 1100 nm located adjacent to the fiber (the period of the interference pattern produced in the fiber core is half that of the phase mask). The gratings had a length of 1 cm. This resulted in TFBGs with a core mode resonance at 1615 nm and several tens of cladding mode resonances in the 1500 to 1600 nm spectral window. A sputtering system (Sky technology development, China) was then used to deposit both a 50 nm-thick gold coating and a 2–3 nm thick chromium underlayer on the fiber surface at the location of the grating. The underlayer improves the surface quality and reliability of the gold coating adhesion to the silica surface of the fiber. It is important to note that the fiber was continually rotated along its axis during the deposition steps to ensure the uniformity of the coating thicknesses around the fiber circumference (see the SEM image of Fig. S22). Thicknesses were further confirmed by ellipsometry measurements (Optical Thickness Meter, Otsuka Electronics). For the tilt angle, since it determines where the maximum of the cladding mode resonances occurs in the spectrum (relative to

the wavelength of the core mode resonance), an angle of 12º was used in order to maximize the amplitude of the resonances in aqueous solutions with refractive indices around 1.32–1.34 (where the SPR active resonances are expected). Finally, in order to avoid having to measure spectra in transmission, which requires the fiber to extend beyond the sensor and to loop back to a spectrum analyzer, the fiber is cleaved several mm downstream from the sensor and the end face coated with a gold layer acting as a back-reflection mirror. This addition allows measurements in reflection (from the single point sensor) since the mirror provides near total broadband reflectivity across the whole spectrum and it provides the added benefit of doubling the amplitude of the sensor response since the light goes through the grating twice. An important additional benefit of the reflective configuration is the elimination of cross-sensitivities of the plasmonic modes to strain on the TFBG since it does not need to be held securely on both sides to remain straight.

### Principle and characteristics of plasmonic fiber-optic sensors.
A Surface Plasmon Polaritons (SPP) is a special kind of electromagnetic wave at visible and near infrared wavelengths that can propagate with a specific phase velocity along the interface between a metallic and a dielectric medium while being confined in the direction perpendicular to the interface. In metal-coated fibers, SPPs can be excited at the outer surface of the metal coating by the evanescent field of a fiber cladding mode under three conditions: (1) the metal layer must be thin enough for the evanescent field of the fiber modes to tunnel from the fiber to the outer metal surface; (2) the phase velocity (as expressed by the effective index) of the fiber mode must be synchronous to that of the SPP wave in order to excite it efficiently; (3) the fiber mode evanescent field must polarized perpendicularly to the metal surface (i.e., radially) because that is the only allowed polarization state for SPPs[55].

The propagation constant of the SPP is given by:

$$\beta_{SPP} = \frac{\omega}{c}\sqrt{\frac{\varepsilon_m \varepsilon_s}{\varepsilon_m + \varepsilon_s}} \quad (1)$$

where c is the speed of light in vacuum, $\omega$ is the angular frequency of light, and $\varepsilon_m$ and $\varepsilon_s$ are the complex relative dielectric constants of the metal and of the dielectric medium adjacent to its surface.

The propagation constants $\beta_{clad,i}$ of cladding modes identified by an arbitrary numerical label (the subscript $i$) are given by:

$$\beta_{clad,i} = 2\pi N_{clad,i}^{eff}/\lambda_{clad,i} \quad (2)$$

where $N_{clad,i}^{eff}$ corresponds to the effective refractive index of $i$-th cladding mode, which is used in the calculation of its resonance wavelength $\lambda_{clad,i}$ from the following phase matching equation of the TFBG:

$$\lambda_{clad,i} = (N_{cald,i}^{eff} + N_{core}^{eff})\Lambda \quad (3)$$

where $N_{core}^{eff}$ is the effective index of the core mode and $\Lambda$ the axial period of the grating.

In fibers with cladding diameter sizes equal to close to 100 times the wavelength, the effective indices (and resonant wavelengths) are very closely spaced because the cladding is a highly multimode guiding structure. Therefore, there are many modes with effective indices sufficiently close to that of the SPP to transfer some light power when the second excitation condition is met, i.e.:

$$\beta_{SPP} = \beta_{clad,i} \quad (4)$$

The consequence of this power transfer is that the amplitudes of the affected cladding mode resonances decreases (as in Fig. 2a). The third condition for efficient SPP excitation further implies that only those modes with electric fields polarized radially at the cladding surface can participate in the power transfer. It turns out that cladding modes excited by TFBGs in single mode fibers come in alternating pairs of radially and azimuthally polarized modes, and that these groups of modes can be excited separately when the light incoming in the fiber core is polarized linearly either in the plane of tilt, or "P-polarized" (to generate radially polarized cladding modes) or perpendicular to it, i.e., "S-polarized" (for azimuthally polarized cladding modes)[56]. These conditions are demonstrated in Fig. S23 where the attenuation of the cladding mode resonances in the SPR spectral region is maximized for P-polarized input light and totally absent for S-polarized input light.

So, in order to perform sensing functions with TFBG-based SPR sensors, Eqs. (1)–(4) provide a direct link between the cladding mode resonance wavelengths where the SPR occurs and the dielectric constant ($\varepsilon_s$) of the medium located just above the metal layer. Changes in the medium properties that impact this dielectric constant will change the value of $\beta_{SPP}$ and hence the wavelengths at which Eqs. (3) and (4) are satisfied. As the envelope of the SPR shifts (to longer wavelengths for instance, when the dielectric constant increases) cladding mode resonances on the short wavelength side of the SPR couple less to the SPP and their amplitude increases as a result (of course, the opposite occurs for negative changes of dielectric constant). This is the sensing methodology used here. The same effect is obtained when a new material layer forms on the metal surface, thus gradually replacing the initial substance that was present there (usually a liquid). This particular modality is widely used in pharmaceutical biomolecular research to measure binding rates and affinity between molecules (attached to the metal surface). In our case, the changes in the complex dielectric constants giving rise to SPR shifts are due to ionic currents and changing charge density in the electrolytes

(above the metal surface) as well as dynamic and static charge re-distributions in the metal itself.

### Preparation of the MO electrode.
Both $MnO_2$ and PEDOT were synthesized by electrodeposition using a CHI-760E electrochemical workstation, a three-electrode system was applied, commercial carbon fiber paper as the working electrode, graphite carbon rod as the counter-electrode, and saturated calomel electrode as the reference electrode. Before electrodeposition, the commercial carbon paper was ultrasonically washed in ethanol for 15 min. The deposition electrolyte of $MnO_2$ was a mixture of 0.12 M $Mn(NO_3)_2$, 0.12 M $NaNO_3$ and 0.04 M sodium dodecyl sulfate. The deposition voltage was 1.5 V, the deposition time was 10 min, and the deposition was made at room temperature. In order to further improve the crystallinity, $MnO_2$ was calcined in air at 300 °C for 1 h after deposition to obtain the MO electrode. Next, we used $MnO_2$ deposited for 10 min as the working electrode to further coat the conductive polymer PEDOT. First, weigh 0.03 M of PEDOT monomer, 0.2 M $LiClO_4$ and 0.07 M of sodium dodecyl sulfate were dissolved in deionized water as the electroplating solution. The electrodeposition of PEDOT was performed by applying 1 V voltage at room temperature for 10 min. Finally, the MO@PEDOT electrode was obtained.

### Preparation of the electrolyte.
All reagents are of analytical grade and are directly used without any purification. 28.7 g of zinc sulfate heptanohydrate and 6.76 g of manganese sulfate monohydrate powder were dissolved in 100 mL of deionized water and stirred by magnetic force for 15 min to ensure their full dissolution. Another electrolyte is obtained by dissolving 6.76 g of monohydrate manganese sulfate powder in 100 mL of deionized water, and the pH is adjusted by $H_2SO_4$ to be the same as the former one.

### Electrochemical measurements.
GCD, CV, PSCA, and EIS were conducted using the Princeton electrochemical workstation (PARSTAT MC) at room temperature. All these measurements were carried out in a three-electrode system, with MO or MO@PEDOT as a working electrode, saturated calomel as reference electrode and graphite rod as counter electrode. For PSCA tests, the electrode was first stabilized after two charging and discharging cycles under GCD tests and then tested using PCSA method with two potential stepping models (from −0.1 V to 0.55 V and from 1 V to 0.45 V). EIS tests were performed as follows: stop the GCD test when the required potential is reached; keep charging or discharging at this specific potential for 10 min to achieve a steady state; carry out EIS test.

### Material characterization.
Field-emission SEM (SEM, JSM-6330F) was used to characterize the microstructure of the MO and MO@PEDOT electrodes. XRD measurements were performed on D-MAX 2200 VPC, RIGAKU using Cu Kα radiation with a 1.5418 Å wavelength at room temperature. The surface chemistry of the electrodes was studied by XPS (NEXSA, Thermo FS) with a mono-chromatized Al Kα radiation (1486.6 eV). Survey and high-resolution spectra were obtained at pass energies of 50 and 30 eV, respectively. The binding energy was calibrated using the C 1s peak (284.8 eV) as reference.

## Data availability
The data that support the findings of this study are available from the corresponding authors upon request.

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

## Acknowledgements

We acknowledge the support of the Key Program of National Natural Science Foundation of China (No. 62035006), the National Natural Science Foundation of China (No. 21822509, No. 22075330, No. 61975068, No. 62011530459), the Guangdong Outstanding Scientific Innovation Foundation (No. 2019TX05×383) and the Program of Marine Economy Development Special Fund under Department of Natural Resources of Guangdong Province (No. GDNRC [2021]33). J.A. acknowledges the support of NSERC (RGPIN-2019-06255).

## Author contributions

T.G. and X.L. conceived the sensors, supervised the project and analyzed the data. R.W., H.Z., Q.L., and X.H. fabricated the materials and carried out experiments. F.L., G.X., and J.A. provided the theoretical analysis. T.G., X.L., J.A., R.W., and H.Z. wrote the paper with the inputs from all authors. K.L. and X. L. helped to reviewed and edited the paper.

## Competing interests

The authors declare no competing interests.
