## [Peer review file · Nature Communications]

REVIEWER COMMENTS

Reviewer #1 (Remarks to the Author):

In this manuscript, Wang et al investigate the operand ionic concentration kinetics at the electrode surface of an aqueous battery via optical fiber sensing based electrochemical surface plasmon spectroscopy (SPR). By the utility and affinity of SPR technique to battery cell application, the authors succeeded to obtain information about ion-absorption/desorption kinetics on working battery electrode. The work may be interesting to researchers on battery fields as well as basic interface electrochemistry. However, some issues should be addressed as follows.

[1] Some similar report has been published by the authors and other group, such as [<https://doi.org/10.1038/s41377-018-0040-y>] and [<https://doi.org/10.1515/nanoph-2019-0504>]. The reviewer recommends adding some explanations about these reports. Especially, difference or progress between above references and this manuscript will be better to discuss.

[2] Generally, SPR technique detect the change of refractive index on sensor-metal surface. Therefore, ion-concentration change on the surface, which caused by ion-absorption/desorption should be described by the refractive index change of electrolyte near the surface. In this point, can the authors show the change of refractive index on the surface by the H⁺ or Zn²⁺ ion concentrating, by some liquid simulation.

[3] The authors tested by constant current charge-discharge and cyclic-voltammetry operation. Please explain the prospect of this technique for other electrochemical mode. I think potential step chronoamperometry (PSCA) test will be better to study the relaxation of ion-concentration near the electrode surface, for instance.

Reviewer #2 (Remarks to the Author):

Operando monitoring of ion activities in rechargeable batteries is an important topic, which demands high-efficient sensors working reliably in the electrochemical reaction condition. The authors targeted at this crucial topic and developed an electrochemical surface plasmon resonance (SPR) optical fiber sensor to monitor the ion intercalation / deintercalation at the electrode-electrolyte interface in aqueous batteries under real working condition. This operando optical fiber sensor finds a stable correlation between the optical signals and the electrochemical results and fills an important gap in current battery monitoring methods. It is practical and meaningful to guide the design and characterization of electrochemical materials and systems. The paper was well-organized and analyzed. Thus, I strongly recommend it for publication after addressing the following concerns.

1) In page 3, a tilting degree of 12° is chosen to “maximize coupling to cladding modes that are suitable to transfer energy to surface plasmons in the aqueous electrolyte of Zn-ion batteries”, stated by the authors. However, there is a lack of evidence of choosing this specific angle. The authors should consider involving the optimization process or provide more explanation.

2) The authors claimed the superiority of this technique over conventional electrochemical techniques, such as cyclic voltammetry or galvanostatic charge-discharge profile. However, the information provided by optic sensing and CV seems to be mostly identical, signified with specific voltage at which certain ion-intercalation occurs. The authors should consider unique advantages of optic sensing, such as batch processing and the mapping capability.

3) There are several cladding modes modulated by the SPR (see the Figure 2d). Why adopted the resonance with the wavelength of 1565.7 nm. Please explain the criterion for dip selection.

4) How would the author remove the cross-sensitivity of such sensor to temperature and strain while the battery is under working? The authors are suggested to provide the details.

5) How fast can the authors collect a spectrum? Can the author realize a portable instrumentation in field measurement (such as the energy storage power stations or electric cars)? It will be very helpful to commercialize such a technique and make it meaningful.

Reviewer #3 (Remarks to the Author):

The results of the manuscript are noteworthy and of high quality. Utilization of tilted FBG SPR sensors with the fiber optic platform combined with core mode reference sensing is a clever and effective means to locally monitor the local chemical environment of the fiber optic sensing probe. The tilted FBG sensors are carefully optimized in order to produce the necessary resonant cladding mode characteristics excited at the same wavelength as the corresponding SPR modes. Careful mounting of the fiber optic sensors at the location of the electrochemical working electrode enables a clear and reversible signal which can be correlated with more traditional electrochemical measurements.

Integration of fiber optic sensors within batteries and energy storage devices is an emerging theme within the science and engineering community, and the local chemical probing is a new area for which very little prior work exists within the community. The topic of the current manuscript is therefore timely and potentially impactful.

Through the derivative of the time varying optical signal, the authors are able to correlate the measurand with expectations for a more traditional cyclic voltammetry experiment. However, the authors do not provide sufficient direct comparison between traditional voltammetry and the obtained results using optical sensing. Such results are highly desirable due to the difference in the sampling associated with both techniques, in which the electrochemical measurements would be a result of the overall electrode reaction while the fiber optic sensor measures only signals in direct proximity (within $\sim 500\text{nm} - 1\text{ micron}$ of the probe surface). In addition, the authors do not provide a clear and strong mechanistic explanation for the correlation between ion rate and the time derivative of the optical signal. Only a brief discussion of this point is found within the main body of the text and these details should be discussed more carefully and explicitly.

In addition, further details of the interaction between the SPR and the tilted FBG excitation of the fiber optic sensing probe would be helpful. The reduced attenuation of the transmitted signal in the vicinity of the SPR is not intuitive and could benefit from additional explanation. Similarly, it is not fully clear what impact on the SPR spectra that the modification to the local index is having. For traditional SPR probes, a simple shift in the SPR wavelength with a slight peak intensity modification would be expected and directly proportional to the local medium refractive index. In many cases a simple refractive index based argument can be made to help explain such observations, is that possible in this case?

Reviewer #1:

In this manuscript, Wang et al investigate the operand ionic concentration kinetics at the electrode surface of an aqueous battery via optical fiber sensing based electrochemical surface plasmon spectroscopy (SPR). By the utility and affinity of SPR technique to battery cell application, the authors succeeded to obtain information about ion-absorption/desorption kinetics on working battery electrode. The work may be interesting to researchers on battery fields as well as basic interface electrochemistry. However, some issues should be addressed as follows.

Technical comments:

[1] Some similar report has been published by the authors and other group, such as [<https://doi.org/10.1038/s41377-018-0040-y>] and [<https://doi.org/10.1515/nanoph-2019-0504>]. The reviewer recommends adding some explanations about these reports. Especially, difference or progress between above references and this manuscript will be better to discuss.

As for the first reference: The first work from our group (named "In situ plasmonic optical fiber detection of the state of charge of supercapacitors for renewable energy storage") aims at quantifying the **STATIC ion concentration** near the electrode surface of supercapacitors. We demonstrate there that such method can be used to infer the state of charge (SOC) of the supercapacitor.

Figure 1. (a) Configuration of ion transport based on grating assisted surface plasmon resonance and (b) corresponding concentration kinetics process.

Compared to this previous work, our current manuscript focuses on understanding the operation mechanism and evaluating the efficiency of aqueous batteries. Benefiting from the extremely high sensitivity (10^{-6} ~ 10^{-8} RIU), fast response time (1 s) and super fine spatial resolution (sub- μm -scale) of the proposed SPR optical fiber sensor, we reveal the correlation

between **ion transport KINETICS** at electrolyte-electrode interface and the optical signal of the fiber sensor. By mapping the change rate of the optical signal with temporal voltage (as the P/V relationship curve shown in above Figure 1), a reversible redox reaction with ion intercalation/deintercalation has been found. Moreover, by analyzing the charging plateau of the P/V relationship curve, we can directly quantify the amount of different ions transport and intercalation/deintercalation. We believe it is a powerful tool for guiding the design of electrode materials and optimizing existing electro-chemistries.

As for the second reference: The second work from Shengchun Liu *et al* (named “Direct detection of charge and discharge process in supercapacitor by fiber-optic LSPR sensors”) [R1], the authors are using cladding-removed multimode fiber and functionalizing the fiber surface with gold nanoparticles for localized SPR generation. It should be noted that cladding removal will highly reduce the mechanical strength of the fiber. Meanwhile, the nanoparticle coatings used in this work are very hard to reproduce for enabling mass production.

On the contrary, our proposed SPR sensor is fabricated in a grating-assisted commercially available “normal” single mode fiber with a uniform gold coating, which therefore ensures the reliability and feasibility (**with no structural modifications**) together with low cost and process availability (the well-established phase masks grating inscription with **standard batch-coating** processes) to **ensure that the devices produced are identical**. It provides sensing inside batteries with highly stable and reproducible measurement.

Figure 2. (a) Localized SPR based on clad removed multimode fiber and (b) SPR based on grating assisted single mode fiber.

Secondly, it should be noted that the spectral width of the LSPR sensor is very broad (larger than 50 nm, as shown in above Figure 2(a)), which limits the measurement accuracy achievable. Our tilted FBG offers an additional resonant mechanism that has the effect of providing high-density narrow spectral combs (with a spectral width of the resonance ~ 0.2 nm, shown in Figure 2(b)) that overlap with the broader absorption of SPR and thus brings the limit of detection for refractive index in the range of 10^{-6} to 10^{-8} RIU (refractive index unit). Therefore, it provides **a unique tool to measure small shifts of the SPR with high accuracy** for highly

sensitive ion transport monitoring.

Finally, compared to the conventional SPR based on ultraviolet–visible light (the second reference), the TFBG-assisted fiber-optic SPR works in the near infrared and therefore provides a much longer penetration depth (more than 500 nm) and propagation length (more than 1 mm) at near infrared wavelengths near 1550 nm. Such propagation lengths cover the most active electron transfer and ion transport area over the electrode surface.

Manuscript modifications: The discussions above and the second reference have been added in the revised manuscript (see the red colored texts in section 2.1, page 2+3, and Ref 46).

[R1] Siyu Qian, Xinlong Chen, Shiyu Jiang, et al, "Direct detection of charge and discharge process in supercapacitor by fiber-optic LSPR sensors," *Nanophotonics* 2020; 9(5): 1071–1079.

[2] Generally, SPR technique detect the change of refractive index on sensor-metal surface. Therefore, ion-concentration change on the surface, which caused by ion-absorption/desorption should be described by the refractive index change of electrolyte near the surface. In this point, can the authors show the change of refractive index on the surface by the H^+ or Zn^{2+} ion concentrating, by some liquid simulations.

Figure 3. Experimental spectra response of TFBG-SPR as a function of H^+ (a) and Zn^{2+} (b) concentration. The Zn^{2+} ion liquid is obtained by the process where 7.2, 14.4, 21.5, 28.7, 35.9 and 43.1 g of zinc sulfate heptahydrate were dissolved in 100 mL of deionized water, respectively, and stirred by magnetic force until dissolved to obtain 0.25, 0.50, 0.75, 1.00, 1.25 and 1.50 M zinc sulfate solutions. The same series of sulfuric acid solutions were prepared with concentrated sulfuric

acid (95.0%-98.0%). (c) the refractive index of H⁺ (Zn²⁺) ion concentration tested by commercial abbe refractometer (Shanghai LICHEN-BX Instrument Technology, WAY-2WAJ, resolution: ± 0.0002 RIU) at the constant temperature of 24 °C, and (d) the corresponding SPR (attenuated cladding mode family in (a) and (b)) wavelength position.

Additional experiments have been carried out and the results reported in Figure 3. A simple linear red shift of SPR hybrid cladding guided mode envelope (Figure 3(a) and (b)) together with the intensity modification for single resonance was tested when external medium ion concentration (H⁺ and Zn²⁺ concentration ranging from 0.25 mol/L to 1.5 mol/L) is increased, which is further supported by measuring their corresponding refractive index in Figure 3(c) (i.e. 1.3355 to 1.3465 for H⁺ ion and 1.3395 to 1.3705 for Zn²⁺ ion, respectively). Apparently, the ion liquid refractive index increases linearly with rising ion concentration, and the SPR guide mode (the attenuated cladding mode in transmission spectra) is linearly sensitive to external refractive index change, and thus can be used to calibrate the ion-concentration change (Figure 3(d)) caused by ion-absorption/desorption. The corresponding refractive index perturbation (ion-concentration) on the fiber surface can be described by wavelength shift of SPR guided mode (for large range of refractive index change) or amplitude modulation (for small range of refractive index change).

Manuscript modifications: The experimental results have been added in the revised supplementary information (see the Figure S8).

[3] *The authors tested by constant current charge-discharge and cyclic-voltammetry operation. Please explain the prospect of this technique for other electrochemical mode. I think potential step chronoamperometry (PSCA) test will be better to study the relaxation of ion-concentration near the electrode surface, for instance.*

Thanks for your academic advice. Accordingly, we further investigated the relationship between the current variation detected by PSCA test and the SPR power change collected by optical fiber, as an application demonstration of this technique for other electrochemical modes to study the relaxation of ion-concentration near the electrode surface. The PSCA tests are carried out using a MO@PEDOT electrode which is in a full-ion-intercalation-state after being stabilized by two charging and discharging cycles under GCD tests. The potential is changed after a standing time of 10 s. Precisely, the potentials are stepped from -0.1 V to 0.55 V (Figure 4(a)) and from 1 V to 0.45 V (Figure 4(b)) to study ion kinetics at charging and discharging states, respectively. Obviously, the potential step at 10 s initiates the direction reversal of the current. Correspondingly, the SPR power, an indicator of ion concentration at electrolyte-electrode interface, increases at 0.55 V (corresponding to ion deintercalation) and decreases at 0.45 V (corresponding to ion intercalation). Moreover, obvious peaks can be observed in d(dB)/dt curves, indicative of the rapid transport rate of the ions at the stepping points, which is consistent with the mechanism of the Zn-ion batteries [R2-R3]. Therefore, this technique can be applied to in situ detection of ion kinetics in various electrochemical modes, which would shine more light on electrochemical mechanism exploration and advanced electrode design in the future. Related results have been added in the revised manuscript.

Figure 4. Potential versus time curve (green), PSCA curve (red), SPR power curve (black) and differential of light power (blue) of MO@PEDOT electrode. The potentials are stepped **a** from -0.1 V to 0.55 V and **b** from 1 V to 0.45 V.

R2. Shen X, Wang X, Zhou Y, et al. Highly Reversible Aqueous Zn-MnO₂ Battery by Supplementing Mn²⁺-Mediated MnO₂ Deposition and Dissolution. *Advanced Functional Materials*, 2021: 2101579. <https://doi.org/10.1002/adfm.202101579>.

R3. Chao D, Zhou W, Ye C, et al. An Electrolytic Zn-MnO₂ Battery for High-Voltage and Scalable Energy Storage. *Angewandte Chemie International Edition*, 2019, 131(23): 7905-7910.

Manuscript modifications: The experimental results have been added in the revised supplementary information (see the Figure S18).

Reviewer #2:

Operando monitoring of ion activities in rechargeable batteries is an important topic, which demands high-efficient sensors working reliably in the electrochemical reaction condition. The authors targeted at this crucial topic and developed an electrochemical surface plasmon resonance (SPR) optical fiber sensor to monitor the ion intercalation / deintercalation at the electrode-electrolyte interface in aqueous batteries under real working condition. This operando optical fiber sensor finds a stable correlation between the optical signals and the electrochemical results and fills an important gap in current battery monitoring methods. It is practical and meaningful to guide the design and characterization of electrochemical materials and systems. The paper was well-organized and analyzed. Thus, I strongly recommend it for publication after addressing the following concerns.

Technical comments:

[1] In page 3, a tilting degree of 12 is chosen to “maximize coupling to cladding modes that are suitable to transfer energy to surface plasmons in the aqueous electrolyte of Zn-ion batteries”, stated by the authors. However, there is a lack of evidence of choosing this specific angle. The authors should consider involving the optimization process or provide more explanation.

Figure 5. The simulation of transmission spectrum of the TFBG device was carried out by first calculating the vector mode fields and effective index of cladding modes as function of tilt angle and wavelength with a cylindrical finite difference mode solver, after which the corresponding spectra (P-polarized input core guided light) were achieved by using complex coupled-mode theory followed by a Runge-Kutta algorithm for the optical propagation through the grating. The properties used for the fiber were: core radius=4.1 μm , cladding radius=62.5 μm , cladding material of pure silica (SiO_2), and core material of germanium-doped silica with 0.0625 germanium/silicon ratio [R5], Au thickness=50 nm (with accepted values for bulk Au thin film refractive indices, including wavelength dispersion [R6]).

The selection of the optimum tilting angle of fiber grating is determined by the refractive index of the electrolyte used in the Zn-ion batteries. The maximum sensitivity is obtained when the cladding modes have effective indices close to the index of the electrolyte which has been measured to be 1.33, with fluctuations of less than 0.02 during charging/discharging cycles. For a grating with a Bragg wavelength near 1620 nm, cladding modes with effective indices near 1.33 occur in the spectral region between 1560 and 1570 nm. In order to maximize the amplitudes of the cladding modes at these wavelengths, the tilt angle was selected based on simulations of TFBG spectra (shown in Figure 5). They confirm that the maximum of the envelope of resonance amplitudes moves towards shorter wavelengths as the tilt angle increases and that the SPR location falls in the correct spectral window [R4].

R4. J. Albert, L.Y. Shao, and C. Caucheteur, "Tilted fiber Bragg grating sensors," *Laser Photonics Reviews*, 7(1), 1-26 (2012).

R5. W. J. Zhou, Y. Zhou, and J. Albert, "A true fiber optic refractometer," *Laser & Photonics Reviews* 11(1), 1600157 (2017).

R6. R. L. Olmon, B. Slovick, T. W. Johnson, D. Shelton, S. H. Oh, G. D. Boreman, and M. B. Raschke, "Optical dielectric function of gold," *Physical Review B* 86, 235147 (2012).

Manuscript modifications: The simulation results have been added in the revised manuscript (see the red colored texts in section 2.1, page 2,) and the supplementary information (see the Figure S4).

[2] *The authors claimed the superiority of this technique over conventional electrochemical techniques, such as cyclic voltammetry or galvanostatic charge-discharge profile. However, the information provided by optic sensing and CV seems to be mostly identical, signified with specific voltage at which certain ion-intercalation occurs. The authors should consider unique advantages of optic sensing, such as batch processing and the mapping capability.*

It is true that the similar information can be extracted from optic sensing and CV since they are both related to electrochemical processes at specific voltages. However, they are two different yet complementary techniques definitely not identical. The CV curves provide the redox potential, current intensity and charge/discharge reaction details which are commonly used for electrode performance quantification, while SPR optical sensing reflects ion transport kinetics and electrode interactions that are indispensable for electrochemical mechanism exploration. That is the reason why the two kinds of signals do not reach maximum values at the same time (Figure 6). More importantly, the CV curve is a result of the overall electrode reaction while the fiber optic sensor measures only signals in direct proximity (within ~500nm to 1 μ m) of the probe surface. So more detailed information could be extracted, with high spatial resolution, from optic sensing. Taking aqueous Zn-ion battery system in our case as an example, CV curves do demonstrate that the electrochemical performance of MO can be improved by PEDOT coating but the fundamental cause of such phenomenon is unclear. By contrast, the semi-quantitative analysis of TFBG-SPR response reveals that the PEDOT layer improves the capacity of MO mainly by promoting the H⁺ diffusion kinetics.

In addition, the optic fiber sensors proposed here are based on well-established

fabrication processes to guarantee the batch processing and mass production. In detail, we use commercially available fibers with no structural modifications, well-established grating inscription processes (the use of phase masks grating based devices instead of interferometric methods), standard coating processes (metal surface coating methods such as electroplating or sputtering) in which multiple devices can be prepared simultaneously, to ensure that the devices produced are identical. By using wavelength division multiplexing (WDM) technology, multiple optical sensors can be cascaded in one optical fiber and be implanted at different positions of one battery, to achieve mapping capability with high time resolution. Finally, it has also been demonstrated that the required instrumentation and protocols are sufficiently simple to be implemented in field (electric vehicles or energy storage stations) use at reasonable cost by operators with minimal training. This ensuring battery reliability, lifetime and sustainability are becoming a possibility.

Therefore, the combination of two techniques will shine more light on the design of electrode materials and optimizing existing electro-chemistries. To highlight the differences between the information provided by optic sensing and CV, additional discussion is added in the revised manuscript.

Figure 6. Spotting voltage-resolved relationship curve of MO and MO@PEDOT electrodes. **a** Derivative of the SPR intensity P'/V and **b** the corresponding CV curve at 1 mV s^{-1} .

Manuscript modifications: The discussions above have been added in the revised manuscript (see the red colored texts in section 2.2, page 7).

[3] There are several cladding modes modulated by the SPR (see the Figure 2d). Why adopted the resonance with the wavelength of 1565.7 nm. Please explain the criterion for dip selection.

The resonance with the wavelength of 1565.7 nm is the first guided mode on the left of the most attenuated resonance. It is the most sensitive mode to external refractive index perturbation (hybrid plasmon guided mode) which is widely used for sensing [R7]. The corresponding simulation is shown in Figure 7(a), in which all attenuated cladding modes covering the plasmon band are labeled from 1 to 10, and the corresponding fitted envelope is shifting to longer wavelength when external refractive index rises. If we perform a first

derivative for the fitted envelope (Figure 7(b)), the cladding mode marked by **number 2** shows the largest slope and therefore the most sensitive to RI changes, especially for very slight surface RI perturbations during the charging/ discharging process inside batteries.

Figure 7. (a) The simulation of a 12° TFBG-SPR as a function of external refractive index (solid line). The corresponding fitting envelope of the attenuated cladding mode resonances covering the plasmon band is moving to longer wavelengths (dash line). (b) the fitted envelope of the SPR (black line) and its 1st derivative (blue line).

R7. V. Voisin, J. Pilate, P. Damman, P. Mégret, and C. Caucheteur, “Highly sensitive detection of molecular interactions with plasmonic optical fiber grating sensors,” *Biosensors & Bioelectronics* 51, 249–254 (2014).

Manuscript modifications: The analysis and figure have been added in the revised the supplementary information (see the Figure S5).

[4] How would the author remove the cross-sensitivity of such sensor to temperature and strain while the battery is under working? The authors are suggested to provide the details.

As for the cross-sensitivity of temperature: it will be perfectly removed temperature crosstalk by considering the RELATIVE wavelength shift of cladding mode to core mode (core mode is independent of external RI perturbation) because they are showing the “same” temperature sensitivity of temperature, shown in the Figure 8. In practical application inside batteries, the temperature variation will be compensated in real time during the data collection of both cladding mode and core mode, by a data processing correction of the cladding mode shifts from that of the core mode shift.

Figure 8. Experimental test of TFBG temperature responses: (a) transmission spectra of 12^o TFBG as a function of external temperature and (b) the temperature sensitivity of core mode (reference mode) and high order cladding mode.

As for the cross-sensitivity of strain: we specially designed a reflected optical fiber sensing probe by coating 100 nm thick Au thin film at the end of the fiber tip (acting as a reflection mirror). The measured “transmission” spectrum of the device is therefore measured in “reflection” from the fiber end. In this situation, the strain is removed because the sensor end is free-standing, inside the batteries as shown in Figure 9. Observations of the core mode stability during the charging / discharging process confirm that the strain is absent from the measurements. Non-axial strains, such as bending, are also not appearing in the data, as such perturbations would only affect low order cladding modes near the Bragg resonance (within 2~4 nm) i.e. far away from plasmon cladding modes (~60 nm far from the core mode) [R8].

Figure 9. (a) Reflection-configuration of TFBG-SPR probe; (b) probe inside the batteries for real time detection of charging and discharging.

R8. T. Guo, L. Shao, H. Y. Tam, P. A. Krug, and J. Albert, “Tilted fiber grating accelerometer incorporating an abrupt biconical taper for cladding to core recoupling,” *Optics Express*, 17, 20651-20660 (2009).

Manuscript modifications: The analysis and figure have been added in the revised the supplementary information (see the Figure S7).

[5] How fast can the authors collect a spectrum? Can the author realize a portable instrumentation in field measurement (such as the energy storage power stations or electric cars)? It will be very helpful to commercialize such a technique and make it meaningful.

It depends on the interrogation system used. For the results presented in this manuscript, a broadband source and optical spectrum analyzer were used. The spectrum analyzer used is capable of acquiring a full spectrum every 3.0 seconds, much faster than the time scale necessary to monitor the battery charging and discharging processes which last more than 6000 seconds. If faster acquisition times were ever needed for this kind of applications, there are other options such as commercially available tunable laser & photodetector combinations which can provide full spectrum acquisition times of the order of a few milliseconds, or customized low-cost options such as the prototypes currently under development in our group (Figure 10) [R9 and R10].

Figure 10. High speed SPR interrogation scheme (up), spectral response and its prototype (bottom).

R9. T. Guo, F. Liu, F. Du, Z. Zhang, C. Li, B. O. Guan, and J. Albert, "VCSEL-powered and polarization-maintaining fiber-optic grating vector rotation sensor," *Optics Express*, 21, 19097-19102 (2013).

R10. Yuke Liu, Binghao Liang, Xuejun Zhang, Nan Hu, Kaiwei Li, Francesco Chiavaioli, Xuchun Gui, Tuan Guo*, "Plasmonic fiber-optic photothermal anemometers with carbon nanotube coatings", *IEEE Journal of Lightwave Technology*, 37, 3373-3380 (2019).

Manuscript modifications: The analysis and figure have been added in the revised manuscript (see the section of Discussion, page 8) and supplementary information (Figure S21).

Reviewer #3:

The results of the manuscript are noteworthy and of high quality. Utilization of tilted FBG SPR sensors with the fiber optic platform combined with core mode reference sensing is a clever and effective means to locally monitor the local chemical environment of the fiber optic sensing probe. The tilted FBG sensors are carefully optimized in order to produce the necessary resonant cladding mode characteristics excited at the same wavelength as the corresponding SPR modes. Careful mounting of the fiber optic sensors at the location of the electrochemical working electrode enables a clear and reversible signal which can be correlated with more traditional electrochemical measurements.

Integration of fiber optic sensors within batteries and energy storage devices is an emerging theme within the science and engineering community, and the local chemical probing is a new area for which very little prior work exists within the community. The topic of the current manuscript is therefore timely and potentially impactful.

[1] Through the derivative of the time varying optical signal, the authors are able to correlate the measurand with expectations for a more traditional cyclic voltammetry experiment. However, the authors do not provide sufficient direct comparison between traditional voltammetry and the obtained results using optical sensing. Such results are highly desirable due to the difference in the sampling associated with both techniques, in which the electrochemical measurements would be a result of the overall electrode reaction while the fiber optic sensor measures only signals in direct proximity (within ~500nm - 1 micron of the probe surface). In addition, the authors do not provide a clear and strong mechanistic explanation for the correlation between ion rate and the time derivative of the optical signal. Only a brief discussion of this point is found within the main body of the text and these details should be discussed more carefully and explicitly.

Figure 11. Spotting voltage-resolved relationship curve of MO and MO@PEDOT electrodes. (a) Derivative of the SPR intensity P'/V and (b) the corresponding CV curve at 1 mV s^{-1} .

The characteristic peaks of traditional voltammetry and optical sensing shift with the same trend (lower charging potential and higher discharge potential), both demonstrating the

two-step ion insertion mechanism and improved battery performance of MO@PEDOT electrode (Figure 11). However, the two techniques focus on different response parameters and thus reveal different information of the electrochemical system. The CV curve performs the current characteristics of the electrodes, and the P'/V curve reflects the change rate of ion concentration at diffusion layer. To be specific, CV curve intuitively provides redox potential, current intensity and charge/discharge reaction details for the electrochemical performance quantification of the electrodes, such as the calculated specific capacity of MO (65 mA h g^{-1}) and MO@PEDOT (174 mA h g^{-1}) electrodes according to the Figure 11(b). In comparison, P'/V curve cannot quantify electrochemical performance but provide a new opportunity to investigate the ion kinetics and electrolyte-electrode interactions by directly monitoring ion transport rate at the interface.

Moreover, the high sensitivity of optic sensing can provide a more obvious characteristic peak to explore the kinetic changes in the charging and discharging processes than CV curve (Figure 12). Benefiting from the extremely high sensitivity ($10^{-6}\sim 10^{-8}$ RIU), fast response time (1 s) and super fine spatial resolution (sub- μm -scale) of the proposed SPR optical fiber sensor, we reveal the correlation between **STATIC ion concentration** & **ion transport KINETICS** at electrolyte-electrode interface and the optical signal of the fiber sensor, which can be used to infer the state of charge (SOC) and quantify the amount of different ions transport and intercalation/deintercalation.

We believe the combination of two techniques (traditional cyclic voltammetry and the proposed optical fiber sensor) is conducive to the evaluation of energy storage performance and in-depth exploration of the electrochemical mechanism, which will afford new probability for guiding the design of electrode materials and optimizing existing electro-chemistries.

Figure 12. Schematic diagram of STATIC and KINETIC state for the relation between optical sensor and electrolyte.

Manuscript modifications: Additional discussions are added in the revised manuscript

to further compare the results obtained by traditional voltammetry and optical sensing (see the red colored texts in section 2.2, page 7)

[2] In addition, further details of the interaction between the SPR and the tilted FBG excitation of the fiber optic sensing probe would be helpful. The reduced attenuation of the transmitted signal in the vicinity of the SPR is not intuitive and could benefit from additional explanation. Similarly, it is not fully clear what impact on the SPR spectra that the modification to the local index is having. For traditional SPR probes, a simple shift in the SPR wavelength with a slight peak intensity modification would be expected and directly proportional to the local medium refractive index. In many cases a simple refractive index based argument can be made to help explain such observations, is that possible in this case?

Figure 13. (a) Gold-coated TFBG with two hybrid resonances ①+② for SPR excitations. (b) Comparison between the best theoretical SPR response for 50 nm gold on silica in the Kretschmann-Raether configuration (thick blue line) and a measured TFBG-SPR spectrum with the same thickness of gold (thin red line) [R11]. The arrows indicate the resonance to be followed in each case, in which the high Q-factor TFBG resonance (0.1 nm) provides a unique tool to measure small shifts of the Plasmon with high accuracy by transforming SPR wavelength shift into amplitude change.

Differing from the traditional bulk prism in the Kretschmann-Raether configuration, a tilted fiber grating assisted SPR sensor provides “TWO hybridized resonances ①+②” as Figure 13(a) shows. By tilting the grating fringes, the core-to-cladding resonant mechanism ① has

the effect of providing a high-density narrowband spectral comb (tens to hundreds of narrowband cladding resonances with individual bandwidth ~ 0.1 nm and Q-factor at the level of 10^4) that overlap with the broader absorption (more than 10 nm wide in the best of cases) of the surface Plasmon ②. As in standard SPR, RI changes produce a shift of the SPR wavelength maximum. The hybridization of the grating resonances with that of the SPR provides a unique tool to measure small shifts of the SPR (due to RI changes) with high accuracy by measuring amplitude changes of the grating resonance located on the edge of the SPR as it shifts, as seen on Figure 13(b). Furthermore, differential measurements between SPR matched cladding modes and those are not matched can be used to further improve measurement accuracy [R12].

Figure 14. The core mode (and ghost modes) is insensitive to surrounding refractive index (SRI) so as to provide an inherent reference to remove any potential influence of temperature and power level fluctuations.

Moreover, another most important feature of the TFBG spectrum is the presence of the Bragg (core mode) resonance that is immune (both in wavelength and power) to the external medium and that can further be advantageously used to de-correlate unwanted temperature and power level fluctuation effects from the sensor response, as shown in Figure 14.

R11. J. Albert, S. Lepinay, C. Caucheteur, and M. C. DeRosa, "High resolution grating-assisted surface plasmon fiber optic aptasensor", *Methods*, 63, 239-254 (2013).

R12. D. Feng, W. Zhou, X. Qiao, and J. Albert, "High resolution fiber optic surface plasmon resonance sensors with single-sided gold coatings," *Optics Express* 24, 16456-16464 (2016).

Manuscript modifications: The analysis and figure have been added in the revised manuscript (see the section 2.1, page 2) and supplementary information (Figure S3).

REVIEWERS' COMMENTS

Reviewer #1 (Remarks to the Author):

The authors well addressed reviewers' comments and certainly revised their manuscripts. Now these materials can publish in current form.

Reviewer #2 (Remarks to the Author):

The authors have well-addressed all comments, the paper can be accepted as it is.

Reviewer #3 (Remarks to the Author):

The authors have adequately addressed the initial review comments.